# A Systematic Review on Attenuation of PCSK9 in Relation to Atherogenesis Biomarkers Associated with Natural Products or Plant Bioactive Compounds in In Vitro Studies: A Critique on the Quality and Imprecision of Studies

**DOI:** 10.3390/ijerph191912878

**Published:** 2022-10-08

**Authors:** Rahayu Zulkapli, Mohd Yusmiaidil Putera Mohd Yusof, Suhaila Abd Muid, Seok Mui Wang, Al’Aina Yuhainis Firus Khan, Hapizah Nawawi

**Affiliations:** 1Institute of Pathology, Laboratory and Forensic Medicine (I-PPerForM), Universiti Teknologi MARA (UiTM), Sungai Buloh Campus, Jalan Hospital, Sungai Buloh 47000, Selangor, Malaysia; 2Faculty of Medicine, Universiti Teknologi MARA (UiTM), Sungai Buloh Campus, Jalan Hospital, Sungai Buloh 47000, Selangor, Malaysia; 3Faculty of Dentistry, Universiti Teknologi MARA (UiTM), Sungai Buloh Campus, Jalan Hospital, Sungai Buloh 47000, Selangor, Malaysia

**Keywords:** PCSK inhibitor, PCSK9, endothelial cells, natural products, atherogenesis, atherosclerosis

## Abstract

A systematic review was performed to identify all the related publications describing PCSK9 and atherogenesis biomarkers attenuation associated with a natural product and plant bioactive compounds in in vitro studies. This review emphasized the imprecision and quality of the included research rather than the detailed reporting of the results. Literature searches were conducted in Scopus, PubMed, and Science Direct from 2003 until 2021, following the Cochrane handbook. The screening of titles, abstracts, and full papers was performed by two independent reviewers, followed by data extraction and validity. Study quality and validity were assessed using the Imprecision Tool, Model, and Marker Validity Assessment that has been developed for basic science studies. A total of 403 articles were identified and 31 of those that met the inclusion criteria were selected. 13 different atherogenesis biomarkers in relation to PCSK9 were found, and the most studied biomarkers are LDLR, SREBP, and HNF1α. In terms of quality, our review suggests that the basic science study in investigating atherogenesis biomarkers is deficient in terms of imprecision and validity.

## 1. Introduction

Systematic reviews in the context of basic research are uncommon. However, despite the rareness, there were systematic reviews of in vitro studies [1,2,3,4]. Systematic reviews for basic science provide the same benefits as those conducted for preclinical animal studies: to statistically combine the results of numerous related studies to provide more reliable results on which decisions can be made and evidence gaps are identified. Basic science can be translated into clinical practice based on solid evidence, and basic research validation is improved by identifying results within multiple model systems [5].

The proprotein convertase subtilisin/Kexin type 9 (PCSK9) has gained attention as a potential therapeutic target for lowering cholesterol levels, especially in homozygous familial hypercholesterolemia (FH)/high-risk and/or category patients who do not reach the low-density lipoprotein (LDL) target, a major risk factor for cardiovascular diseases [6,7,8]. The discovery of the 9th or the last member of the protein convertase family known as PCSK9 was reported in 2003 by Nabil Seidah [9]. Until it was discovered, there were only two known genes (*LDL-R and ApoB*) related to FH in humans [10]. The classical method of action involves PCSK9 protein chaperoning the low-density lipoprotein receptor (LDLR) to intracellular degradative organelles, hence accelerating its degradation [11]. The consequent reduction in surface LDLR impedes LDL clearance, yielding an increase in plasma LDL cholesterol (LDL-C). The discovery of PCSK9 took a sharp turn in the lipid field with PCSK9 inhibitors becoming an undeniable therapeutic reality. Mice and humans without functional PCSK9 appear healthy [12,13], and it seems that therapeutic inhibition of PCSK9 unlikely would have any serious adverse effects. This makes PCSK9 a very promising potential therapeutic target for dyslipidemia therapy.

Currently, numerous prospective medications that inhibit the PCSK9 pathway have entered preclinical or early phase clinical trials, and the FDA has approved two of these treatments (evolocumab and alirocumab) [14,15]. According to preclinical research, PCSK9 has pleiotropic effects beyond regulating plasma LDL-C levels and may be a crucial factor in the pathogenesis of atherosclerosis [16,17]. The PCSK9 inhibition attenuates atherosclerosis progression and lowers the risk for acute cardiovascular events [6,18]. PCSK9 inhibition may be best achieved by identifying and developing small compounds that may be taken orally and have anti-PCSK9 action. The history of pharmacology has offered compelling evidence on the significance of identifying naturally occurring substances with potential therapeutic actions, and the in vitro studies have provided persuasive evidence of the relevance through molecular mechanisms [19]. The atherogenic inhibition by the natural products in in vitro studies was conducted by measuring the expression of the inflammatory, adhesion molecules, oxidative stress, endothelial nitric oxide synthase (eNOS), and nuclear factor-κB (NF-kB) biomarkers [20,21,22].

Therefore, this review aimed to gather, compare and critique the imprecision and quality of the in vitro research that is published on bioactive compounds or natural-product-derived PCSK9 inhibitors involving PCSK9 and atherogenic biomarkers inhibition rather than the detailed reporting of the results evidence.

## 2. Methods

The literature search and systematic review methods adhered to the Cochrane Collaboration guidance [23] to reduce the risk of bias and error. This review allows the Preferred Reporting Items for Systematic Reviews (PRISMA) guidelines (Appendix A) [24].

### 2.1. Definitions

PCSK9 inhibition was defined as the hindrance of PCSK9 molecule binding to e LDLR, so that the LDLR degradation can be prevented, thus increasing LDLR being recycled to the surface of hepatocytes for LDLC uptake, and reducing blood LDLC level. Atherogenesis biomarkers are either protein or gene expression that was affected by atherosclerosis. Natural products were defined as substances or chemicals produced by plants. Plant bioactive compounds referred to a type of chemical found in small amounts in plants.

### 2.2. Search Criteria

Electronic literature searches in the Scopus, PubMed, and Science Direct databases was conducted between 2003 and 2021. The starting year is following the year of *PCSK9 gene* discovery as the third gene linked to autosomal dominant hypercholesterolemia [25]. Search strategies are presented in Appendix B. The selected databases were searched on 27 August 2021 up to 30 August 2021. The publications found using the keyword combinations of ‘Proprotein Convertase Subtilisin Kexin 9 Inhibitor* or PCSK inhibitor*’, ‘cell*’, ‘endothelial cell*’ were included. The clinical, diagnostic, or prognostic outcomes were excluded from the review. The time filter used was from 2003 until 2021 to limit the years of publication search.

### 2.3. Inclusion and Exclusion Criteria

The included studies were the original publications of biomarker expression, either protein and/or gene expression of *PCSK9*, and atherogenesis in in vitro studies. The PCSK9 biomarkers were specifically selected and included. The studies of atherogenesis biomarkers without PCSK9 were excluded. The significance and relevance of the selected literature were evaluated based on their content and type of publication. The studies were excluded if (i) the study used other types of PCSK such as PCSK1 or PCSK8, (ii) the study used PCSK9 to observe effects other than lipid-lowering in other diseases, (iii) human subjects or animals were involved, (iv) they were not written in the English language; and (v) the articles were reviews, commentaries, editorials, unpublished manuscripts, or conference abstracts. The non-English research articles, conference proceedings, abstracts, book chapters, and commentaries were not included. All review articles that included clinical, diagnostic, or prognostic outcomes were excluded.

### 2.4. Study Identification and Selection

After identifying articles in the databases mentioned above, these articles were imported into EndNote X20software (Thompson Reuters, Philadelphia, PA, USA), and duplicate articles were removed. The eligibility criteria was used to conduct the first-level screening of articles using titles and abstracts. Full-text articles were then accessed to determine eligible articles to include in the review. A data extraction form was performed to extract study characteristics, including author(s), year of publication, cell lines used, tested plant bioactive compound (PBC) or natural product (NP), biomarkers measured, and expression at protein and gene levels. The titles and abstracts were independently screened by two authors (R.Z. and A.Y.F.K.).

### 2.5. Data Synthesis

A summary of all the included studies was compiled. The data were sorted according to the cell lines, treatment, and the expression of protein and gene levels. The article that discussed more than one cell line or NP/PBC will be separated into different studies (Table 1). Results were presented alongside overall judgments for concerns regarding the validity and imprecision of the result. The data were extracted by a single review author (R.Z.). To ensure accuracy, another review author (A.Y.F.K.) went through the data independently, and any discrepancies were resolved by the third review author (H.N.).

### 2.6. Quality Assessment

The current review emphasized the imprecision and quality of the included research rather than the detailed reporting of the results.

The only risk of bias for non-clinical research is the SYRCLE checklist [26]. In the SYRCLE checklist, the judgment of the domains is either unclear (UNR) or not applicable (NA). SYRCLE is based on the risk of bias tools developed for randomized controlled clinical trials [24]. However, we found that these tools were not appropriate for the design of basic scientific studies. The SYRCLE signaling questions are not relevant to basic science studies, they do not use a language that is meaningful to laboratory scientists, and they do not critique all issues pertinent to the biases of fundamental research. No formal randomization or allocation concealment or blinding is used in laboratory-based research. In addition, every effort is taken to ensure that the experiment and controls are treated equally throughout the study.

Thus, the validity will follow an “Imprecision Tool and Assessment” (Appendix C), “Model Validity Assessment” (Appendix D), and “Marker Validity Assessment” (Appendix E) established by Collins et al. [1] to judge the choice and validation in basic science studies. In “Imprecision Tool and Assessment”, the determination involves a minimum requirement for low risk was that the authors reported technical repeats, interassay repeats, and variability.

## 3. Results

### 3.1. Literature Searches and Inclusion Assessment

A summary of the identification and selection of studies for inclusion in this review is presented in Figure 1, in accordance with the PRISMA statement [24]. Literature searches of electronic databases retrieved 8403 research articles. After the duplicated research articles were removed, 8057 titles/abstracts were screened, and 6791 research articles were excluded as having no relevance to the review. Full research articles of 537 potentially relevant references were selected for further examination. Of these, 505 research articles were excluded after reading the entire article; the reasons for exclusion are provided in Figure 1. Thirty-one publications met the inclusion criteria.

### 3.2. PCSK9 in Relation to Atherogenesis Biomarkers

#### 3.2.1. In Vitro Models

Five different cell lines were identified in the in vitro studies that measured the PCSK9 expression. One study reported PCSK9 attenuation in Human Umbilical Vein Endothelial Cells (*HUVEC*), two in Human Hepatocytes (*Huh 7*), twenty-seven in Human hepatoma (*HepG2*), and one in JLM3 (hepatocellular carcinoma cells) (Table 1). Most studies used hepatocytes cell lines, in accordance with the fact that PCSK9 is highly expressed in the liver [58]. Apart from that, PCSK9 is also present in the kidneys, intestines, brain, and blood vessels [59].

Four in vitro models were identified in all selected studies; (i) Oxidized LDL (Ox-LDL) stimulated cells, (ii) Lipopolysaccharides (LPS) stimulated cells, (iii) Lipoprotein-depleted serum (LDPS) cell growth medium, and (iv) Delipidated-serum (DLPS) cell growth medium. Most of the in vitro research selected in the studies used the LPDS model, and the Ox-LDL model was the least used.

#### 3.2.2. Protein and Gene Expression of PCSK9 In Vitro Models

Using systematic review methodology, we identified thirty-two studies describing PCSK9 expression in relation to thirteen different biomarkers studied in human cells line in terms of protein and gene expression, which were treated by different natural products or plant bioactive compounds. All the natural products or compounds in the selected studies possessed downregulated effects of PCSK9 except for red yeast rice (monacolin K) (Table 1).

##### PCSK9 in Relation to LDLR, SREBP, and HNF1α Biomarkers

Twenty-five studies on PCSK9 measured the LDLR expression. From that, all studies showed the inverse relationship between PCSK9 and LDLR levels. However, six studies reported “not significantly changed/unaffected/no changed” in LDLR gene expression even though in the protein expression, it was highly expressed. Second, thirteen studies reported the PCSK9 and SREBP (Sterol regulatory element-binding proteins). From that, seven studies reported the downregulation of SREBP together with PCSK9. Contrarily, two studies reported that SREBP was upregulated when PCSK9 was downregulated, and the other four studies reported “not significantly changed/not affected” on SREBP when PCSK9 was downregulated. Ten studies discussed the HNF1α biomarker in relation to PCKS9; 8 were downregulated with PCSK9 suppression, 1 was upregulated and 1 was not affected by PCSK9 downregulation (Table 1).

These biomarkers were the least biomarkers investigated in the included articles. Four studies investigated the 3-Hydroxy-3-Methylglutaryl-CoA Reductase (HMGCR) biomarkers; only one study reported the protein and gene expression of HMGCR, while the other studies only reported on the mRNA expression. Three studies reported a direct relationship between HMGCR and PCSK9 mRNA, while forkhead box O3 (FoxO3) biomarkers were upregulated in all four studies. The peroxisome proliferator activated Receptor Gamma (PPARg) protein and gene expression was investigated in two studies, and it was unaffected in both. The inverse relationship between PPARg and PCSK9 gene expression was discovered. Lectin-like oxLDL 1 (LOX-1), NADPH Oxidase 4 (NOX-4), adhesion, and inflammatory biomarkers were only reported in one study included. The biomarkers were downregulated only when PCSK9 biomarkers were downregulated. While for fas cell surface death receptor (FAS), only the gene expression was reported, and they were downregulated when PCSK9 was downregulated (Table 1).

#### 3.2.3. Imprecision and Validity Analysis

The imprecision tool for basic science studies was created with the purpose of judging how well the authors reported sample size, statistical methodology, and variability (2). The minimum requirement for low risk is for the authors to have well-reported technical and inter-assay repeats as well as variability. Imprecision Tool Assessment (Figure 2) regarded twenty-seven studies (84%) as ‘low concern’ with low ‘technical reporting and statistical rating’, but the sample size rating was unclear. Another seven studies (22%) were regarded as unclear in all domains during the Imprecision Tool Assessment (Table 2, Figure 2). The imprecision of the included articles was evaluated to be unclear in overall rating when: (1) they scored unclear more in one imprecision domain, (2) the number of technical repeats was not mentioned in the article, and (3) the statistical test rating was reported as unclear because no analysis was reported on the comparison.

A model validity tool performed in basic science is to judge how well the authors reported the details and validity of the model used in the research. Assessment of model validity (Figure 2) indicated that most of the studies (66%) were judged to be valid, ten studies (31%) were unclear, and one was considered to be ‘high concern’; the main reasons lie in the ‘no reported’ model for the experiment.

The marker validity analysis focuses on the most studied biomarkers in the included articles (PCSK9 in relation to LDLR, SREBP, and HNF1α). Analysis of the marker validity for PCSK9 showed eighteen studies (56%) scored ‘low’, while the other fourteen studies were judged to be ‘unclear’ (44%). For LDLR, thirteen studies (52%) were evaluated as ‘low’, and eleven studies were ‘unclear’ (44%). One reported ‘high’ due to the absence of positive and negative control. While SREBP and HNF1α biomarkers were judged to be ‘unclear’ in the majority (92% and 80%) of the included studies for marker validity due to the absence of positive control (Table 2).

## 4. Discussion

Both the mRNA and protein levels of gene expression are controlled by on/off switches and fine-tuned regulation [60]. There has been a flurry of research into the connection between mRNA and protein levels across genes, with sometimes contradicting findings [58]. In yeast, the amount of mRNA present can be used as a reliable predictor of the amount of protein present [59]. On the other hand, in mammalian cells, the association has been demonstrated to be much lower and varies considerably depending on the cell type and state. For cells that have been exposed to a stimulus, the situation gets even more complex. When mammalian cells were exposed to protein misfolding stress, the link between protein and mRNA quantities was broken down, and substantial regulation occurred at both the mRNA and protein levels [61]. Thus, it is crucial to evaluate the quality of the research conducted on the biomarkers specifically in atherogenesis as small changes to the protein and mRNA levels affected the outcome.

To the best of our knowledge, this is the first systematic review that describes the PCSK9 in relation to atherogenesis biomarkers that emphasized the imprecision and quality of the research. A gain-of-function mutation in the PCSK9 gene was found to cause FH [62]. The inhibition of PCSK9 attenuates atherosclerosis progression and reduces the risk for acute cardiovascular events [6,18].

The imprecision analysis, model validity, and marker validity have been performed following the basic science study (2). However, some of the exclusion has been made to suit this study. The sample size rating or evaluation included in the imprecision assessment is not relevant to cell studies as the calculation of sample size is unnecessary before conducting the experiment. In cell studies, triplicates were considered enough when the variation was small. This is agreeable with the majority of the selected and evaluated publications that used technical triplicates in their experiment. Thus, the exclusion of sample size rating is appropriate for the overall imprecision score evaluation. In addition, observer variability (technical reporting domain) also is irrelevant to cell studies research as it requires the paper to report whether the experiment gives the same result when repeated. None of the articles reported on the consistency of the results. Statistical analysis is common and good enough in cell studies to observe variation and consistency. Thus, the observer variability was excluded for the overall score of technical reporting. Other than that, overall, none of the manuscripts describes the routine maintenance of the model (domain four) nor the routine checking for the absence of mycoplasma or contaminants (domain seven). It was a crucial practice and routine in cell culture studies; however, it was rarely reported in the manuscript. The experiment’s success is the actual indicator that the routine was performed. Thus, it was unnecessary to report on that. Therefore, the overall rating was made by excluding the score in domains four and seven. The paper that was regarded as ‘high concern’ or ‘high risk’ is the paper that gave no, not applicable, and not reported for all domains 1 to 9.

All the natural products or compounds in the included studies showed the downregulation of PCSK9 except for red yeast rice (monacolin K). Red yeast rice reported the upregulation of PCSK9 upon treatment with HepG2 (24 h). All included studies showed the inverse relationship between PCSK9 and LDLR levels. This supports the theory that PCSK9-bound-LDLR causes the increase in LDLR degradation that impedes LDLC lowering of PCSK9 by direct binding to the epidermal growth factor repeat A (EGF-A) of the LDLR and shuttling the LDLR from the endosomes to the lysosomes for degradation [63].

SREBP controls the genes involved in fatty acid production (SREBP-1c) and cholesterol metabolism, principally regulating PCSK9 at the transcriptional level (SREBP-2) [64]. The PCSK9 gene minimal promoter region contains a sterol regulatory element (SRE) [65]. Nuclear SREBP expression significantly increases PCSK9 promoter activity, and PCSK9 expression can be controlled by nutritional status via a mechanism involving SREBP-1c [66]. For SREBP, the relationship between PCSK9 and SREBP was contradicted in the included studies; (i) SREBP was upregulated when the PCSK9 was downregulated, and (ii) SREBP “not significantly changed/not affected” when PCSK9 was downregulated. The marker validity was reported as ‘unclear’ for the articles that reported “SREBP was reported not significantly changed nor affected’. Besides SREBP2, HNF1α is a critical transcription factor that regulates PCSK9 gene transcription [65]. Most of the studies showed the downregulation of HNF1α with PCSK9 suppression aggregable with the HNF1α function that promotes PCSK9 transcription by binding with the HNF1 motif, which is located upstream of SRE1 in the PCSK9 promoter [67]. Despite the consensus on the outcome of SREBP and HNF1α in relation to PCSK9, the majority of marker analyses for both were regarded as ‘unclear’ due to the absence of positive control. Very few studies (8% and 20%) reported the positive control of SREBP AND HNF1α biomarkers. The reporting of positive controls should be fundamental in basic science study allowing researchers to validate the outcome of their research.

## 5. Conclusions

Cell lines have long been regarded as a valuable resource for basic research as well as pre-clinical studies. Living cells can be used to investigate the functional significance of genetic products such as mRNA, miRNA, and proteins, and cell lines are a valuable research resource. Studying cell lines is also important in investigating a particular medicine’s detailed mechanism or pathway. Even though selection pressures can compromise the predictive value of cell lines during the formation and long-term passaging processes, a significant advantage of cell lines is that examinations can be conducted with high throughput and at a relatively low cost.

Using a systematic review, the relation of PCSK9 with thirteen different biomarkers in different cell lines has been identified. Despite the exclusion of some criteria domain in the validity and imprecision of the included research, the quality of some studies is still questionable. This might be caused by several factors, especially the cost for basic research to be precise and valid. Improvements are still needed in evaluating the validity and imprecision of basic science studies. The establishment of imprecision and validity for a different scope of basic research, particularly in vitro studies, is crucial as it will allow more rapid development of new alternative treatments.

## Figures and Tables

**Figure 1 ijerph-19-12878-f001:**
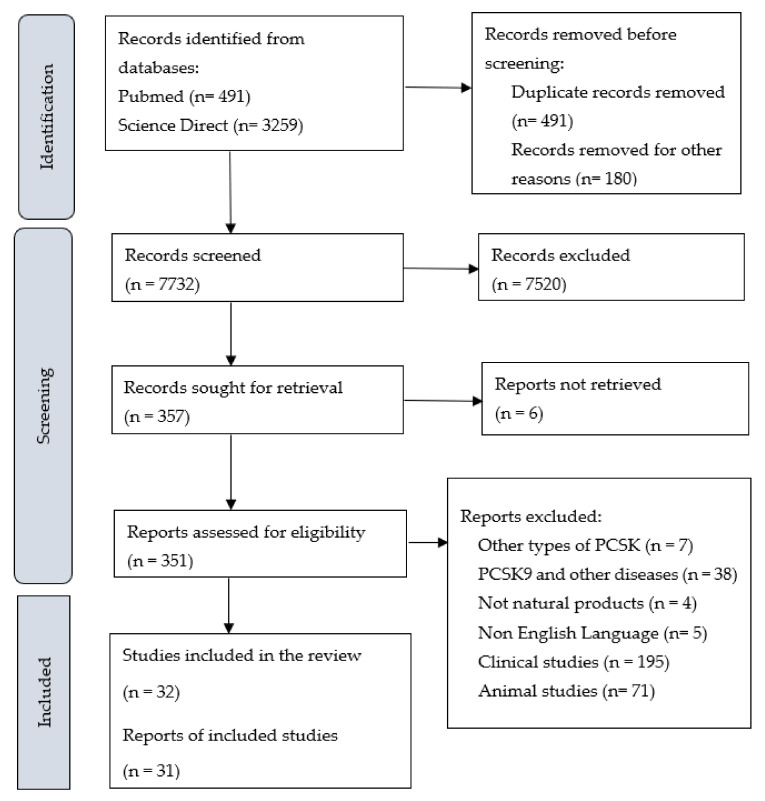
PRISMA Flowchart of Studies.

**Figure 2 ijerph-19-12878-f002:**
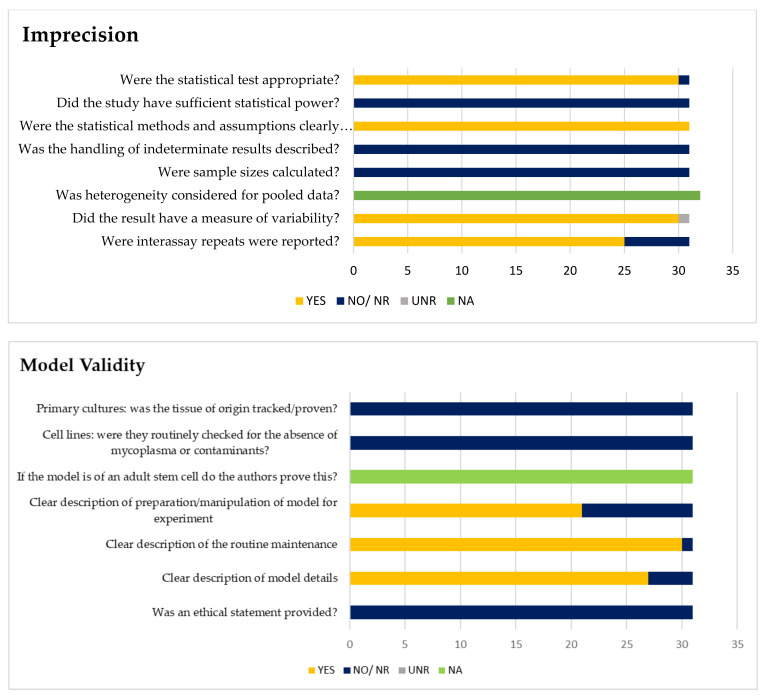
Assessments of imprecision and model validity. Yellow bars = number of studies for judgments of ‘yes’. Dark blue bars = number of studies for judgements of ‘no’ or ‘not reported’. Light grey bars = number of studies ‘unclear (UNR)’ for question (unclear for imprecision). Green bars = number of studies for judgements of ‘not applicable (NA)’.

**Table 1 ijerph-19-12878-t001:** Summary of biomarkers expression of selected studies.

Cell Lines	Study ID	Natural Product/ Plant Bioactive Compound	Biomarkers	Expression at Effective Concentration	
				Proteins	Genes
*HUVEC*	Wang 2019 [27]	Ginkgolide B **	PCSK9	Downregulated	Downregulated
			LDLR	Upregulated	Upregulated
			ICAM-1	Downregulated	Downregulated
			VCAM-1	Downregulated	Downregulated
			SREBP2	Downregulated	Downregulated
			IL-1α	Downregulated	Downregulated
			IL-1β	Downregulated	Downregulated
			IL-6	Downregulated	Downregulated
			MCP-1	Downregulated	Downregulated
			CXCL-1	Downregulated	Downregulated
			CXCL-2	Downregulated	Downregulated
			NOX-4	Downregulated	Downregulated
			LOX-1	Downregulated	Downregulated
*Huh7*	Mbikay 2014 [28]	Quercetin-3-glucoside **	PCSK9	Downregulated	Downregulated
			LDLR	Upregulated	Upregulated
			SREBP2	Not reported	Not affected
	Wang 2020 [29]	Ascorbic acid **	PCSK9	Downregulated	Downregulated
			LDLR	Upregulated	Upregulated
			PPARg	Not affected	Not affected
			FoxO3a	Upregulated	Upregulated
*LO2*	Jing 2019 [30]	Resveratrol **	PCSK9	Downregulated	Downregulated
			LDLR	Upregulated	Upregulated
			SREBP 1c	Downregulated	Downregulated
*HepG2*	Aggrey 2019 [31]	3R3,14-dihydroangustoline **	PCSK9	Downregulated	Not reported
			LDLR	Upregulated	Not reported
	Ahn 2019 [32]	Erybraedin D **	PCSK9	Downregulated	Downregulated
	Cameron 2008 [33]	Berberine **	PCSK9	Downregulated	Downregulated
	Chae 2018 [34]	Saucinone **	PCSK9	Not reported	Downregulated
			LDLR	Not reported	Upregulated
	Chen 2016 [35]	Tanshinone IIA **	PCSK9	Downregulated	Downregulated
			LDLR	Upregulated	NSC
	Choi 2017 [36]	*Allium fistulosum* L. *	PCSK9	Downregulated	Downregulated
			LDLR	Downregulated	Downregulated
			SREBP2	Downregulated	Downregulated
			HNF1α	Not affected	Downregulated
	Dong 2019 [37]	Siblinin A **	PCSK9	Downregulated	Downregulated
*HepG2*	Fan 2021 [38]	Berberine derivative (9k) **	PCSK9	Downregulated	Not reported
			LDLR	Upregulated	Not reported
	Gao 2018 [39]	Pinostrobin **	PCSK9	Downregulated	Downregulated
			LDLR	Upregulated	NSC
			SREBP2	NSC	Not reported
			HNF1α	NSC	Not reported
			FoxO3a	Upregulated	Not reported
	Fu 2020 [40]	17β-estradiol (βE2) **	PCSK9	Downregulated	Not reported
			LDLR	Upregulated	No changed
	Gu 2017 [41]	Lunasin **	PCSK9	Downregulated	Downregulated
			LDLR	Upregulated	Upregulated
			HNF1α	Not reported	Downregulated
			SREBP2	Upregulated	Upregulated
	Hwang 2020 [42]	Butein **	PCSK9	Downregulated	Downregulated
			LDLR	Upregulated	Upregulated
			HNF1α	Downregulated	Downregulated
			SREBP2	NSC	Downregulated
			HMGCR	Not reported	Downregulated
	Hwang 2021 [43]	*Capsella**b**ursa-**p**astoris* *	PCSK9	Downregulated	Downregulated
			LDLR	Not affected	Downregulated
			HNF1α	Downregulated	Downregulated
			SREBP2	Downregulated	Downregulated
	Kim 2020 [44]	Piceatannol **	PCSK9	Downregulated	Downregulated
			LDLR	Upregulated	Not affected
			HNF1α	Not reported	Downregulated
			SREBP2	Not reported	Downregulated
	Lammi 2019 [45]	Lupin peptide T9 **	PCSK9	Downregulated	Not reported
			LDLR	Upregulated	Not reported
			HNF1α	Downregulated	Not reported
	Li 2020 [46]	23,24-Dihydrocucurbitacin B **	PCSK9	Downregulated	Downregulated
			LDLR	Upregulated	Upregulated
			SREBP2	Upregulated	Not reported
			HNF1α	Downregulated	Not reported
	Masagalli 2021 [47]	Moracin C **	PCSK9	Downregulated	Downregulated
	Pel 2020 [48]	5,6,7,4’-tetramethoxyflavanone **	PCSK9	Downregulated	Downregulated
			LDLR	Upregulated	NSC
			HNF1α	Not reported	Downregulated
	Pel 2017 [49]	(+)-pinoresinol **	PCSK9	Downregulated	Downregulated
	Weng 2021 [50]	Gynostemma pentaphyllum *	PCSK9	Downregulated	Downregulated
*HepG2*	Wang 2021 [51]	Gypenoside LVI **	PCSK9	Downregulated	Downregulated
			LDLR	Not reported	Not affected
			SREBP2	Not affected	Not affected
	Wu 2019 [52]	Tetrahydroprotoberberi-ne derivatives **	PCSK9	Downregulated	Downregulated
			LDLR	Upregulated	Not reported
	Wu 2021 [53]	Diallyl disulfide **	PCSK9	Downregulated	Downregulated
			LDLR	Upregulated	Upregulated
			SREBP2	Downregulated	Downregulated
			HMGCR	Downregulated	Downregulated
			HNF1α	Not affected	Not affected
	Yang 2018 [54]	Liraglutide **	PCSK9	Downregulated	Downregulated
			HNF1α	Downregulated	Downregulated
	Yang 2018 [55]	Chitosan oligosaccharides **	PCSK9	Downregulated	Downregulated
			SREBP2	Upregulated	Upregulated
			HNF1α	Upregulated	Upregulated
			FoxO3a	Upregulated	Upregulated
	Lupo 2019 [56]	Monacolin K **	PCSK9	Upregulated	Upregulated
			LDLR	Upregulated	Not reported
			HMGCR	Not reported	Upregulated
			FAS	Not reported	Upregulated
		Berberine **	PCSK9	Downregulated	Downregulated
			LDLR	Upregulated	Not reported
			HMGCR	Not reported	Downregulated
			FAS	Not reported	Downregulated
		1-deoxynojirimycin **	PCSK9	Downregulated	Downregulated
			LDLR	Upregulated	Not reported
			HMGCR	Not reported	Downregulated
			FAS	Not reported	Downregulated
	Wang 2020 [29]	Ascorbic acid **	PCSK9	Downregulated	Downregulated
			LDLR	Upregulated	Upregulated
			PPARg	Not affected	Not affected
			FoxO3a	Upregulated	Upregulated
*JLM3*	He 2017 [57]	*Actinidia chinensis* *	PCSK9	Not reported	Upregulated
			LDLR	Not reported	Upregulated

Abbreviation: HUVEC (Human Umbilical Vein Endothelial Cells); HUH7 (Human Hepatocytes); JLM3 (hepatocellular carcinoma cells); LO2 (hepatocytes); HepG2 (Human Hepatoma); NSC (not significantly changed); * Natural product; ** Plant bioactive compound. PCSK9 in Relation to FoxO3, HMGCR, PPARg, FAS, LOX-1, NOX-4, Adhesion, and Inflammatory Biomarkers.

**Table 2 ijerph-19-12878-t002:** Characteristic, model validity and imprecision of selected studies on the atherogenesis biomarkers.

Biomarkers	Cell Lines	Study ID	Model Validity	Imprecision	Biomarker Validity
PCSK9	HUVEC	Wang 2019 [27]	Low	Low	Unclear
	Huh7	Mbikay 2014 [28]	Low	Low	Low
		Wang 2020 [29]	Low	Low	Low
	LO2	Jing 2019 [30]	Low	Low	Unclear
	JLM3	He 2017 [57]	Unclear	Low	Low
	HepG2	Aggrey 2019 [31]	Unclear	Unclear	Unclear
		Ahn 2019 [32]	Unclear	Low	Unclear
		Cameron 2008 [33]	Low	Low	Low
		Chae 2018 [34]	Unclear	Low	Low
		Chen 2016 [35]	Low	Low	Unclear
		Choi et 2017 [36]	Low	Unclear	Low
		Dong 2019 [37]	Unclear	Low	Unclear
		Fan 2021 [38]	Low	Low	Unclear
		Gao 2018 [39]	Low	Low	Low
		Fu 2020 [40]	Unclear	Low	Unclear
		Gu 2017 [41]	Low	Low	Unclear
		Hwang 2020 [42]	Low	Low	Unclear
		Hwang 2021 [43]	Low	Unclear	Low
		Kim 2020 [44]	Low	Low	Low
		Lammi 2019 [45]	Low	Low	Unclear
		Li 2020 [46]	Low	Low	Unclear
		Masagalli 2021 [47]	Low	Low	Unclear
		Pel 2020 [48]	Unclear	Unclear	Unclear
		Pel 2017 [49]	High	Unclear	Low
		Weng 2021 [50]	Low	Unclear	Low
		Wang 2020 [51]	Low	Low	Unclear
		Wu 2019 [52]	Unclear	Low	Low
		Wu 2021 [53]	Low	Low	Unclear
		Yang 2018 [54]	Low	Low	Unclear
		Yang 2018 [55]	Low	Low	Unclear
		Lupo 2019 [56]	Unclear	Low	Low
		Wang 2020 [29]	Low	Low	Low
LDLR	HUVEC	Wang 2019 [27]	Low	Low	High
	Huh7	Mbikay 2014 [28]	Low	Low	Low
		Wang 2020 [29]	Low	Low	Low
	LO2	Jing 2019 [30]	Low	Unclear	Unclear
	JLM3	He 2017 [57]	Unclear	Low	Low
	HepG2	Aggrey 2019 [31]	Unclear	Unclear	Unclear
		Cameron 2008 [33]	Low	Low	Low
		Chae 2018 [34]	Unclear	Low	Low
		Chen 2016 [35]	Low	Low	Unclear
		Choi 2017 [36]	Low	Unclear	Unclear
		Fan 2021 [38]	Low	Low	Unclear
		Gao 2018 [39]	Low	Low	Low
		Fu 2020 [40]	Unclear	Low	Unclear
		Gu 2017 [41]	Low	Low	Unclear
		Hwang 2020 [42]	Low	Low	Unclear
		Hwang 2021 [43]	Low	Unclear	Low
		Kim 2020 [44]	Low	Low	Low
		Lammi 2019 [45]	Low	Low	Unclear
		Li 2020 [46]	Low	Low	Low
		Pel 2017 [49]	High	Unclear	Unclear
		Wang 2021 [51]	Low	Low	Unclear
		Wu 2019 [52]	Unclear	Low	Low
		Wu 2021 [53]	Low	Low	Unclear
		Lupo 2019 [56]	Unclear	Low	Low
		Wang 2020 [29]	Low	Low	Low
SREBP	HUVEC	Wang 2019 [27]	Low	Low	Unclear
	Huh7	Mbikay 2014 [28]	Low	Low	Unclear
	LO2	Jing 2019 [30]	Low	Unclear	Unclear
	HepG2	Choi et 2017 [36]	Low	Unclear	Unclear
		Gao 2018 [39]	Low	Low	Unclear
		Gu 2017 [41]	Low	Low	Unclear
		Hwang 2020 [42]	Low	Low	Unclear
		Hwang 2021 [43]	Low	Unclear	Low
		Kim 2020 [44]	Low	Low	Unclear
		Li 2020 [46]	Low	Low	Unclear
		Wang 2021 [51]	Low	Low	Unclear
		Wu 2021 [53]	Low	Low	Unclear
		Yang 2018 [55]	Low	Low	Unclear
HNF1α	HepG2	Choi 2017 [36]	Low	Unclear	Unclear
		Gao 2018 [39]	Low	Low	Unclear
		Gu 2017 [41]	Low	Low	Unclear
		Hwang 2020 [42]	Low	Low	Unclear
		Hwang 2021 [43]	Low	Unclear	Low
		Kim 2020 [44]	Low	Low	Unclear
		Li 2020 [46]	Low	Low	Low
		Pel 2020 [48]	Unclear	Unclear	Unclear
		Wu 2021 [53]	Low	Low	Unclear
		Yang 2018 [55]	Low	Low	Unclear
FoxO3a	Huh7	Wang 2020 [29]	Low	Low	Low
	HepG2	Gao 2018 [39]	Low	Low	Unclear
		Yang 2018 [55]	Low	Low	Unclear
		Wang 2020 [29]	Low	Low	Low
HMGCR	HepG2	Hwang 2020 [42]	Low	Low	Unclear
		Wu 2021 [53]	Low	Low	Unclear
		Lupo 2019 [56]	Unclear	Low	Low
PPARg	Huh7	Wang 2020 [29]	Low	Low	Unclear
	HepG2	Wang 2020 [29]	Low	Low	Unclear
FAS	HepG2	Lupo 2019 [56]	Unclear	Low	Unclear
LOX-1	HUVEC	Wang 2019 [27]	Low	Low	Unclear
NOX-4	HUVEC	Wang 2019 [27]	Low	Low	Unclear
ICAM	HUVEC	Wang 2019 [27]	Low	Low	Unclear
VCAM	HUVEC	Wang 2019 [27]	Low	Low	Unclear
(IL)-1α,	HUVEC	Wang 2019 [27]	Low	Low	Unclear
IL-1β	HUVEC	Wang 2019 [27]	Low	Low	Unclear
IL-6	HUVEC	Wang 2019 [27]	Low	Low	Unclear
MCP-1	HUVEC	Wang 2019 [27]	Low	Low	Unclear
CXCL-1	HUVEC	Wang 2019 [27]	Low	Low	Unclear
CXCL-2	HUVEC	Wang 2019 [27]	Low	Low	Unclear

Imprecision interpretation: Low = no concern, Unclear = not enough information to make judgement, High risk = there is a concern of high risk. Model validity interpretation: Low = all domains clearly reported, and there were no additional concerns. Unclear = Any domain was unclear, but not high risk. High risk = there is a concern of high risk.

## Data Availability

Data is available in a publicly accessible repository.

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
