# Peer review of "A Systematic Review on Attenuation of PCSK9 in Relation to Atherogenesis Biomarkers Associated with Natural Products or Plant Bioactive Compounds in In Vitro Studies: A Critique on the Quality and Imprecision of Studies"

_ijerph, 2022, doi:10.3390/ijerph191912878_

Round 1
Reviewer 1 Report (Previous Reviewer 1)
The authors have gone to great lengths to modify this previously submitted manuscript (ijerph-1865081) according to recommendations from prior peer review rounds. I appreciate the authors' point-by-point response and recommend publication.
This manuscript is a resubmission of an earlier submission. The following is a list of the peer review reports and author responses from that submission.
Round 1
Reviewer 1 Report
In their manuscript, Zulkapli and colleagues evaluate in vitro studies of PCSK9 attenuation and associated biomarkers of atherogenesis by natural products or bioactive compounds. The authors have gone to great lengths to screen a large number of studies in various databases. They show that a total of 31 studies have reported on the attenuation of PCSK9 and atherogenesis biomarkers associated with natural products or compounds in an in vitro setting to date. Furthermore, the imprecision and validity of those studies is reported.
Overall, this systematic review has several major flaws that are an impediment to publication in its current form.
Major concerns:
1. The whole manuscript lacks substance and is incoherent in terms of reported objectives and what is actually presented.
2. Natural products or bioactive compounds needs to be defined. It is currently hard to understand which molecules are classified as either a natural product or a bioactive compound by the authors. Why were Evolocumab and Alirocumab, which are both monoclonal antibodies, included as key words in the literature search? This needs to be justified.
3. A PRISMA checklist needs to be published in the appendix.
4. Line 124-125: This needs to be clearly stated in the title, abstract, and introduction/objectives.
5. It appears the article has been submitted using the template of a different journal (International Journal of Molecular Sciences) beginning page 5. The template of International Journal of Environmental Research and Public Health must be used for submission in this journal.
6. The results section is lacking a detailed analysis of natural products and compounds used in the included in vitro studies.
7. Discussion section 3.1 (i.e., the entire discussion) should be moved to results section. A discussion is essentially missing.
8. Similar to the discussion, the conclusion is misleading and does not reflect what is suggested by the title, abstract, and introduction.
Minor concerns:
1. While certainly interesting, a two-paragraph introduction of the concept of systematic reviews, in general, and its application in preclinical studies, in particular, is excessive. To better tailor this introduction to the needs of the informed readership, it should be condensed to a maximum of 1 paragraph.
2. Line 64-65: Why is lowering PCSK9 mRNA highlighted here as an example to reduce PCSK9?
3. Line 65-66: What does this sentence mean?
4. Line 107-108: This is a redundant sentence. Article types excluded are also listed in line 112-113.
5. Table 2 in its current form is very confusing and way too long (page 5-12). It should be either reformatted, shortened, or put into the appendix.
Reviewer 2 Report
Dear authors
1- please explain the main conclusion of your study in more details in the abstract, the introduction of the abstract can be shortened
2-the section 2.1 should contain all definitions or should be merged to the other sections, it is very short and unnecessary
3-type of the paper criteria should be screened in the initial phases, not at the end
4-Figure 2 is not clear and should be replaced with a paragraph or another type of presentation
5- Discussion part should be initiated with the main findings and then discuss it further, you may categorize it into subsections after the main explanation, but needs more discussion in each part
6- the conclusion should be concise and shortened to the main conclusion based on your findings solely
Round 2
Reviewer 1 Report
Dear authors,
While clearly efforts have been made to improve the overall quality of the manuscript, additional major revisions are required to meet a standard that provides benefit to the scientific community. This does not only include moving around sentences, adding new sentences here and there, reformatting figures, etc. - but actually improving the scientific merit of the manuscript. The authors are kindly referred to my original point-by-point description of major concerns.
Reviewer 2 Report
Dear Authors
Thank you for the reply
1-tables 1 and 2 can be shortened or depict as a graph, as it can be easily understood,
2- the present table 1, last row has no title and not comprehensive